# Enhancing Pediatric Outpatient Medical Services Through the Implementation of the Smart Well Child Center Application

**DOI:** 10.3390/healthcare13141676

**Published:** 2025-07-11

**Authors:** Naporn Uengarporn, Teerapat Saengthongpitag, Poonyanuch Chongjaroenjai, Atcha Pongpitakdamrong, Wutthipong Sriratthnarak, Phonpimon Rianteerasak, Kanyarat Mongkolkul, Paninun Srinuchasart, Panuwat Srichaisawat, Nicharee Mungklang, Raiwada Sanguantrakul, Pattama Tongdee, Wichulada Kiatmongkol, Boonyanulak Sihaklang, Piraporn Putrakul, Niwatchai Namvichaisirikul, Patrapon Saritshasombat

**Affiliations:** 1School of Pediatrics, Institute of Medicine, Suranaree University of Technology, Nakhon Ratchasima 30000, Thailand; atcha.po@sut.ac.th (A.P.); wutthipong@sut.ac.th (W.S.); phonpimon.ri@sut.ac.th (P.R.); kanyarat.mo@sut.ac.th (K.M.); paninun.sr@sut.ac.th (P.S.); panuwat.s@sut.ac.th (P.S.); nicharee@sut.ac.th (N.M.); raiwada@sut.ac.th (R.S.); wichulada.ki@sut.ac.th (W.K.); boonyanurak@sut.ac.th (B.S.); pirapornput@sut.ac.th (P.P.); 2Suranaree University of Technology Hospital, Suranaree University of Technology, Nakhon Ratchasima 30000, Thailand; teerapat.sa@g.sut.ac.th (T.S.); poonyanuch123@gmail.com (P.C.); 3School of Obstetrics and Gynecology, Institute of Medicine, Suranaree University of Technology, Nakhon Ratchasima 30000, Thailand; pattama_t@sut.ac.th; 4School of Family Medicine and Community Medicine, Institute of Medicine, Suranaree University of Technology, Nakhon Ratchasima 30000, Thailand; niwatchai@sut.ac.th; 5Institute of Nursing, Suranaree University of Technology, Nakhon Ratchasima 30000, Thailand; patrasa@sut.ac.th

**Keywords:** pediatric outpatient department, Well Child Center, application development

## Abstract

Background: Caregivers of children often encounter barriers when accessing pediatric healthcare services. These challenges highlight the need for digital innovations to improve accessibility and efficiency in pediatric outpatient care. Objectives: This study aimed to design, implement, and pilot evaluate the Smart Well Child Center application in conjunction with enhancements to the Pediatric Outpatient Department. Methods: This study employs a mixed-methods research approach. The application was developed following the system development life cycle (SDLC) process, and its performance was subsequently evaluated. Additionally, its effectiveness in real-world settings was assessed through a satisfaction survey completed by 85 child caregivers. The results were summarized using the mean and standard deviation, and satisfaction levels were compared using paired *t*-test and repeated measures ANOVA. Results: The findings reveal that caregivers face significant challenges, including financial burdens related to travel, prolonged wait times, and difficulties accessing healthcare services. In response, the application was designed to incorporate key functionalities. Within the pre-consultation self-assessment module, caregivers can complete evaluations and receive recommendations directly through the application. Furthermore, the service procedure flowchart was restructured to seamlessly integrate these digital innovations, thereby enhancing the overall healthcare experience. The evaluation results indicate that the application achieved high performance ratings across all assessed dimensions (4.06 ± 0.77). Additionally, caregivers reported a substantial increase in satisfaction levels both immediately after implementation (4.58 ± 0.57) and one month afterward (4.59 ± 0.33). Conclusions: Given these findings, it is recommended that the hospital fully adopt the Smart Well Child Center application to improve healthcare accessibility and reduce patient wait times. Future research should assess the long-term impact of the intervention on both caregiver outcomes and healthcare professional workflow, satisfaction, and system usability, to inform broader implementation strategies.

## 1. Introduction

Early childhood represents a critical period for growth, development, and the establishment of lifelong health behaviors. Well-child care (WCC) services, delivered through pediatric outpatient departments (OPDs), play an essential role in promoting children’s physical, emotional, and developmental health, while also contributing to the reduction of health disparities [1,2]. In Thailand, WCC services are primarily provided through public health facilities under the Ministry of Public Health, including sub-district health promotion hospitals, community hospitals, and regional hospitals. These services follow national Maternal and Child Health (MCH) guidelines and include growth monitoring, immunization, developmental screening (e.g., Denver II or DSPM), nutrition counseling, and anticipatory guidance. Thailand’s universal health coverage (UHC) ensures these services are free for all children under five. Understanding the structure, accessibility, and caregiver engagement within this system is essential for evaluating its impact. The development of comprehensive WCC models, particularly those enhancing parental health literacy, has the potential to significantly improve child-rearing practices and long-term child outcomes [3,4]. Previous research has identified persistent challenges to effective pediatric service delivery—such as long wait times, limited appointment slots, geographic isolation, transport-related costs, and inadequate developmental guidance [5,6]. While these findings stem mainly from high-income settings, similar issues are observed in Thailand, especially in rural areas, where caregiver knowledge gaps and language barriers among ethnic minorities persist. In response, the integration of medical and dental care—referred to as medical–dental integration—has been increasingly advocated as a strategy to enhance disease prevention efforts in pediatric populations. Empirical studies investigating the association between children’s attendance at well-child visits (WCVs) and their receipt of preventive dental visits (PDVs) have demonstrated that WCV attendance is positively correlated with an increased likelihood of obtaining preventive dental care [7]. Still, the transferability of these findings to regions with different health system infrastructures, availability of pediatric dental services, or caregiver literacy levels must be critically assessed.

To address these challenges, a range of healthcare innovations, including telemedicine, mobile health applications, and electronic service-learning models, have been explored. Some evidence suggests that parental satisfaction with both the process and quality of care tends to increase when parents are actively involved in collaboration with healthcare professionals [8]. A pilot study conducted in the United States found that telemedicine follow-up for infants discharged from the neonatal intensive care unit (NICU) was well-accepted by parents and contributed to a reduction in emergency department visits and in-person consultations [9]. Furthermore, the use of eHealth technologies for self-management after hospital discharge—particularly following pediatric surgery or preterm birth—has shown promising outcomes among users, especially those with prior experience in digital health platforms [10]. Additional studies report that virtual home visits increase assessment completion, reduce visit time, and enhance child–caregiver–staff relationships [11]. Still, regional implementation must consider limitations such as internet access, digital literacy, and resource disparities [12]. In parallel, eHealth applications have been developed to support child care, exemplified by the Child Care Service application, which offers a secure and user-friendly platform for searching, booking, payment processing, and real-time communication and information sharing between parents and caregivers. Real-time online programs have similarly been associated with improved breastfeeding practices, enhanced maternal self-efficacy, earlier initiation of breastfeeding within the first hour after birth, and prolonged exclusive breastfeeding durations [13,14]. Moreover, the utilization of maternal and child health hotline services staffed by specialized nurses has been well-accepted and has shown potential in improving service accessibility and the quality of life for mothers and children within community settings [15]. Clinic websites and telephone consultations have also been positively received in Thai community health centers [5,16]. Hybrid telehealth models—blending synchronous and asynchronous formats—have shown efficacy in identifying children at risk for language delays [17]. As such, regionally relevant digital interventions should focus on addressing service gaps, promoting equity, and ensuring cultural appropriateness. The ongoing development of child care applications has consistently demonstrated high user satisfaction and alignment with caregiver needs [18].

Despite the increasing adoption of health information technologies, previous interventions have predominantly emphasized either remote telehealth consultations or isolated eHealth education platforms, without fully integrating these services into a cohesive ecosystem that spans the pre-visit, visit, and post-visit experiences. Furthermore, limited research has evaluated the effectiveness of such integrated systems, particularly in the context of pediatric outpatient service redesign within middle-income countries such as Thailand. A critical gap persists in the availability of application-based pediatric service models that simultaneously address service accessibility, caregiver education, developmental screening, and caregiver satisfaction through a unified digital platform implemented within a real-world hospital setting in Thailand. In addition, significant concerns remain regarding the safety of telehealth utilization and the appropriateness of its application in certain clinical scenarios, both of which have not been thoroughly investigated. In response to these gaps, the present study aims to develop and implement the Smart Well Child Center application, integrating it into the pediatric outpatient workflow, evaluating its effectiveness in enhancing caregiver satisfaction with pediatric healthcare services at Suranaree University of Technology Hospital. The intervention also seeks to reduce outpatient service time, promote the physical, emotional, and developmental health of children, and contribute to the reduction in health inequalities.

## 2. Materials and Methods

### 2.1. Study Design

This research employed a mixed-methods design, integrating application development and pilot testing within a real-world healthcare setting. The study was conducted at the Pediatric Outpatient Department of Suranaree University of Technology Hospital (SUTH), Thailand. The overall project timeline spans from July 2023 to January 2025. However, the results presented in this manuscript reflect an interim analysis based on data collected during the pilot implementation phase, which took place from September to November 2024. In the development phase, the application was developed following the system development life cycle (SDLC) model [19], comprising the following four phases:

(1)Planning phase: The planning phase spanned approximately four months and involved a structured, multi-stakeholder collaboration process. In-depth discussions were conducted with pediatricians, nurses, hospital administrators, and IT personnel to evaluate the existing WCC service workflow. These consultations aimed to: (1) assess operational feasibility for digital integration; (2) map current service processes and identify gaps; (3) analyze pain points affecting caregiver satisfaction and service time; (4) gather functional and technical requirements for the proposed application; and (5) align clinical goals with system design priorities. This phase also included informal caregiver interviews to understand user expectations and digital literacy challenges. Outputs from these discussions were translated into a set of system specifications, which were used to design low-fidelity wireframes and flow diagrams for stakeholder feedback. Parallel to this, the IT team conducted an infrastructure readiness assessment to evaluate compatibility with existing hospital systems.(2)Analysis phase: A comprehensive analysis of the current workflow and the proposed new workflow was conducted, with particular emphasis on identified problems and user needs. This phase also involved the establishment of a MySQL database management system, enabling data exchange between the WCC application database and the HosXP system in the hospital information system (HIS) via an API.(3)Design phase: The physical service area was developed alongside the application. User interfaces and data visualizations were designed for various user groups, including administrators, physicians, nurses, and child caregivers.(4)Implementation phase: The application was deployed in the production environment and underwent extensive user testing over the course of one month. During this period, user feedback was systematically collected to identify and resolve issues. The application’s performance was evaluated by both IT professionals and end-users.

The application was developed in compliance with institutional data security protocols to protect the privacy of all users, particularly given the sensitivity of pediatric health information. Data entered through the application were encrypted using Secure Sockets Layer (SSL) protocols during transmission and stored in a secure MySQL database. Access to identifiable patient data was restricted to authorized personnel through multi-level authentication. The application also required caregiver login with individual credentials to prevent unauthorized access.

### 2.2. Participants

A total of 85 child caregivers participated in this study. Participants were recruited from the pediatric outpatient department at SUTH using a simple random sampling technique. Eligible participants included primary caregivers aged 18 years and older who were responsible for children under 6 years of age and who had accessed the Smart Well Child Center services at least once during the study period. Caregivers were excluded if they had communication barriers (e.g., language difficulties, hearing impairments without support) or if they declined to provide informed consent. Recruitment was conducted across multiple clinic sessions throughout the week to help ensure sample diversity and minimize selection bias.

### 2.3. Outcome Measures

The Smart Well Child Center application was developed with four core modules: appointment scheduling; pre-consultation self-assessments and immediate child care advice integrated into the application after meeting the physician via the SUTH form; chatbot-delivered health education; and telemedicine consultation. To complement these digital services, the physical layout of the outpatient department was restructured into functional zones. The performance of the application was evaluated by three domain experts across four principal dimensions: functional requirements, functionality, usability, and security. In addition, its effectiveness in real-world clinical settings was assessed based on feedback from 85 child caregivers using six evaluation criteria: accessibility of services, quality of medical care, timeliness of service delivery, adequacy of medical equipment and general environment, performance of Smart Well Child features, and overall caregiver satisfaction [20]. Both evaluation forms underwent quality assessment, which included expert review for content validity and calculation of the index of consistency (IOC). The evaluation forms were then piloted and tested for reliability using Cronbach’s alpha, yielding reliability scores of 0.95 and 0.92, respectively.

### 2.4. Data Analysis

The results were summarized using means and standard deviations. Comparisons of satisfaction levels were conducted using paired *t*-tests and repeated measures 
analysis of variance. The evaluation of the Smart Well Child Center application’s performance and user satisfaction was classified into five levels based on 
mean scores:

Mean 4.21–5.00: Performance and satisfaction were interpreted as being at the highest level.

Mean 3.41–4.20: Performance and satisfaction were interpreted as being at a high level.

Mean 2.61–3.40: Performance and satisfaction were interpreted as being at a moderate level.

Mean 1.81–2.60: Performance and satisfaction were interpreted as being at a low level.

Mean 1.00–1.80: Performance and satisfaction were interpreted as being at the lowest level.

### 2.5. Ethical Considerations

The study was approved by the Suranaree University of Technology Ethics Committee (EC66-51), date of approval 26 July 2023 to 25 July 2025. All participants were informed of the study’s objectives, and informed consent was obtained.

## 3. Results

The study sample received services at the Pediatric Outpatient Department. Caregivers identified several critical challenges, including financial burdens related to travel, prolonged wait times, and difficulties in accessing healthcare services. In response to these issues, the Smart Well Child Center application was developed with core functionalities, and the physical layout of the service area was restructured to support the seamless integration of these digital innovations. The research findings consist of the following.

### 3.1. Development of the Smart Well Child Center Application and Integration into Clinical Service Infrastructure

The development of the Smart Well Child Center application integrates several key functionalities aimed at enhancing service efficiency and caregiver convenience. Core features include an appointment scheduling system that enables child caregivers to independently confirm appointments, check in for services, reschedule, or cancel appointments through the application. Within the pre-consultation self-assessments module, caregivers are able to complete evaluations and obtain immediate feedback while awaiting service, thereby contributing to a reduction in overall service time. Moreover, the application provides chatbot-based health education and telemedicine services, facilitating direct communication between caregivers and healthcare professionals via a LINE chatbot. Through this platform, caregivers can seek medical consultations, submit inquiries, access nutritional information, initiate telephone consultations, and utilize telemedicine services. The user interface of the application is illustrated in Figure 1.

In parallel with the development of the application, the physical service area and service delivery system were redesigned to support the integration of digital technologies. The restructured layout includes: (1) a screening station, (2) a data entry station within the application to streamline caregiver interaction, (3) a growth assessment station, (4) a developmental assessment station, (5) a dental examination station, (6) a history-taking station, (7) a physician consultation station, (8) a telehealth/telemedicine station via the LINE platform, (9) a vaccination station, and (10) a counseling and appointment scheduling station. At each point, caregivers may circulate to receive assessments based on the service context. If the queues at stations 2 through 5 are unoccupied, caregivers can proceed to complete the assessments. Each service point is flexibly accessible depending on the service context and availability. This dynamic queuing approach aligns with the principles of the simulation-based complexity of healthcare model, which emphasizes adaptive, context-sensitive scheduling to optimize resource utilization and maintain service continuity [21]. The flowchart of the service procedures is depicted in Figure 2.

### 3.2. Pilot Evaluation of the Smart Well Child Center: System Performance and Caregiver Satisfaction

Table 1 shows personal information of child caregivers and the children under their care. Most primary caregivers were female (82.4%), with 77.6% identified as the children’s mothers. Regarding employment, 28.2% were employed as company or store staff. A significant proportion resided in Nakhon Ratchasima province (91.8%). Additionally, 70.6% of caregivers have accessed the service more than three times. Among the children receiving services, a majority were male (63.5%). In terms of physical status following WHO Multicenter Growth Reference, 62.4% had a normal weight, while 23.5% were classified as obese. Regarding height, 63.5% of the children were of normal stature [22]. Developmental screening by the *Developmental Surveillance and Promotion Manual* (DSPM) indicated that 70.6% of the children used baby bottles. Assessments of gross motor skills, fine motor skills, emotional and social development, and language abilities revealed age-appropriate performance across these domains [23]. Furthermore, the majority of children exhibited normal findings during dental check-ups.

Table 2 presents the performance evaluation results of the Smart Well Child Center application across four domains: function requirement test, function test, usability test, and security test. The function requirement test recorded the highest mean score (4.27 ± 0.24), categorized at the “Highest” performance level. This outcome suggests that the application successfully fulfilled its predefined functional requirements to an exceptional degree. The function test achieved a mean score of 4.11 ± 0.31, while the usability test and security test obtained mean scores of 4.00 ± 0.42 and 3.92 ± 0.27, respectively. All three domains were classified as exhibiting ”High” performance levels. These findings indicate that the application not only performs reliably but also demonstrates strong usability and maintains a high standard of security. Overall, the evaluation results affirm that the Smart Well Child Center application upholds high standards across all assessed domains, with particularly notable excellence in meeting functional requirements.

Table 3 presents the satisfaction levels of child caregivers across various healthcare service domains assessed before the implementation of the Smart Well Child Center application, immediately after its use, and one month following its use. The findings reveal statistically significant improvements across all domains (*p* < 0.01). Satisfaction with ease of access to services increased from a moderate level (mean = 3.29 ± 0.56) prior to use to the highest level immediately after use (4.58 ± 1.10) and sustained one month later (4.60 ± 0.40). Similarly, satisfaction with the quality of medical care improved markedly from a moderate level (3.27 ± 0.42) to the highest level post-intervention (4.57 ± 0.51) and after one month (4.66 ± 0.52). Service timeliness also showed notable enhancement, with mean scores rising from 3.26 ± 0.69 before use to 4.55 ± 0.58 immediately after use and 4.54 ± 0.71 one month later. Satisfaction regarding medical equipment and the general environment increased from a high level (3.58 ± 0.50) to the highest level after use (4.52 ± 0.56) and after one month (4.58 ± 0.58). For the newly introduced Smart Well Child service, satisfaction scores were consistently high following use (4.54 ± 0.55) and one month afterward (4.56 ± 0.59). Furthermore, overall satisfaction demonstrated a substantial improvement, rising from a moderate level (3.35 ± 0.41) before use to the highest level after use (4.58 ± 0.57) and maintaining that level after one month (4.59 ± 0.33). In addition to the quantitative findings, informal feedback collected during follow-up indicated that caregivers particularly appreciated the ability to manage appointments independently and receive instant health-related advice via the chatbot. Several caregivers noted that the pre-consultation self-assessment reduced anxiety before seeing the physician, as it helped them reflect on their child’s condition and feel more prepared. Others mentioned that teleconsultation features were especially helpful for reducing unnecessary travel and wait times.

## 4. Discussion

This study developed the Smart Well Child Center application in parallel with enhancements to the Pediatric Outpatient Department at Suranaree University of Technology Hospital. The application integrates several key features. Within the pre-consultation self-assessment module, caregivers are able to complete evaluations and receive immediate feedback while awaiting services, thereby reducing overall service time. Concurrently, the physical service area and the service delivery system were redesigned to support the integration of digital technologies. Notably, the integration of medical and dental care commonly referred to as medical–dental integration has been increasingly advocated as a strategy to enhance disease prevention efforts in pediatric populations. Previous findings indicated that preventive dental visit attendance is positively correlated with an increased likelihood of receiving preventive dental care [7].

The findings of this study suggest that the development and implementation of the Smart Well Child Center application was associated with perceived improvements in healthcare service delivery and higher reported caregiver satisfaction. These results are consistent with the existing literature, which highlights the potential of telehealth innovations to support pediatric healthcare services. In alignment with a systematic review conducted by Sharma (2023) [24], which reported high levels of family acceptance and comparable clinical outcomes between telehealth and traditional in-person services, the present study similarly observed a marked increase in caregiver satisfaction immediately following the implementation of the application and sustained at one-month follow-up [4,12]. In addition to the quantitative results, informal feedback from caregivers provided valuable insights into the application’s real-world utility. Caregivers expressed appreciation for features that allowed them to manage appointments independently and receive timely health information via the chatbot. The pre-consultation self-assessment was reported to reduce anxiety and enhance preparedness before physician visits, while the teleconsultation functions were viewed as convenient tools for minimizing travel and waiting times. These qualitative impressions further indicate the application’s acceptability and potential usefulness in everyday clinical use. Moreover, the integration of appointment scheduling, pre-consultation self-assessments, and chatbot-based telemedicine services appeared to help address previously reported barriers, such as prolonged wait times and limited healthcare accessibility, thereby supporting evidence from earlier studies that telehealth interventions may improve service access and caregiver experiences [2,11]. Within the broader context of well-child care (WCC) services, the use of telephone consultations and the dissemination of health information through clinic websites have been positively received by healthcare personnel and administrators in community health centers [5,16,25]. Similarly, the deployment of maternal and child health hotline services, staffed by specialized nurses, has demonstrated significant potential in improving service accessibility and enhancing the quality of life for mothers and children [15,26]. Nevertheless, previous studies have emphasized that the effectiveness of these services is highly dependent on the communication skills and specialized training of service personnel. Furthermore, the restructuring of the physical service area to facilitate the integration of digital innovations is congruent with findings from earlier studies, which suggest that telehealth-supported postnatal care can maintain clinical effectiveness while simultaneously reducing the necessity for in-person visits [27].

Similarly, in the present study, the implementation of digital pre-assessment via the SUTH form and scheduling tools was associated with perceived improvements in service timeliness, completeness of assessments, and overall caregiver satisfaction. Due to the inherent complexity of healthcare systems, this study is designed to organize healthcare service delivery using a simulation modeling approach. Simulation models are increasingly recognized as a valuable method for addressing challenges related to appointment scheduling and service efficiency. The complexity of healthcare stems from its intricate structure, which includes patient flows, social interactions, decision-making processes, and queues—often caused by the limited availability of healthcare resources. Modeling these systems at the individual level, rather than at the population level, can yield more accurate and practical insights [28]. A previous study found that in terms of care ratings, over 90% of respondents rated their clinician as either excellent or good. However, when it came to standard anticipatory guidance topics, more than half of parents with children aged 0 to 3 years reported that they had never discussed any age-appropriate topics with a clinician. This highlights the growing importance of anticipatory guidance [29]. To address this gap, we designed the Smart Well Child SUTH form for appointed patents with three steps: pre-doctor self-assessments for caregivers, doctor note, and post-doctor information return for care givers in the SUTH application following an anticipatory guidance and questionnaire form—based on the American Academy of Pediatrics (AAP) guidelines and a consensus among pediatricians to ensure that essential topics are covered during key developmental stages [30]. This form was adapted for ease of use by Thai caregivers, many of whom face barriers related to health literacy and limited digital experience.

The high functionality ratings observed across the function requirement test, function test, usability test, and security test are consistent with previous studies emphasizing that the successful integration of eHealth technologies depends on both user acceptance and technical robustness [9]. The present study supports this assertion, demonstrating that the Smart Well Child Center application was perceived as both technically sound and highly acceptable to caregivers. Furthermore, the development of comprehensive well-child clinic service models, alongside initiatives to promote parental health literacy, may support parental knowledge and skills—critical factors for fostering appropriate child-rearing practices and supporting optimal child development trajectories [3]. However, in line with previous observations, it is essential to recognize that although digital interventions have the potential to improve access and caregiver satisfaction, sustained attention to issues such as digital literacy, user engagement, and equitable access remains vital [12,31]. Additionally, this study has certain methodological limitations that should be acknowledged. First, the absence of a control group limits the ability to draw causal inferences regarding the observed improvements. The changes in caregiver satisfaction and service efficiency, while promising, cannot be conclusively attributed to the intervention alone. Second, the relatively short follow-up period—limited to one month after implementation—does not allow for evaluation of long-term outcomes or sustained user engagement. These limitations should be addressed in future studies to strengthen the evidence base for this digital service model.

Overall, this study adds to the growing body of evidence supporting the role of telehealth in pediatric outpatient services. It illustrates how a well-designed digital application, coupled with a restructured service environment, can effectively overcome traditional healthcare barriers while maintaining high levels of caregiver satisfaction [17,18,32]. In studies related to child nutrition and development, issues such as poor nutrition and developmental delays are commonly observed, with a higher prevalence of obesity compared to underweight children [33]. The global prevalence of being overweight and obese among preschool children has increased. In Thailand, findings from the Fourth National Health Examination Survey (NHES IV) conducted during 2008–2009 reported that 8.5% of preschool-aged children were classified as obese and 3.5% as overweight. Among school-aged children, 10.9% were obese and 4.0% overweight. These national figures are consistent with the trends reported in this study [34]. Additionally, stunting—or growth failure—remains a significant concern and necessitates nutritional guidance beginning at birth. Developmental screening using the *Developmental Surveillance and Promotion Manual* (DSPM) identified delays during the child’s visit. In accordance with national guidelines, appropriate recommendations were provided, and a follow-up appointment was scheduled to reassess the child’s developmental progress at the next visit [23]. To address these issues, it is crucial to offer age-appropriate developmental stimulation, nutritional advice from the earliest stages of life, ensuring that children receive a balanced diet rich in essential nutrients, and to support development through appropriate stimulation tailored to each developmental stage. These efforts help promote healthy growth and overall child development. Providing guidance to parents on both nutrition management and developmental support is essential at every stage of a child’s life. These needs reflect broader national trends and highlight the importance of coordinated efforts by government agencies and relevant stakeholders. To ensure effectiveness, such support should include mechanisms for performance reporting, data-driven decision-making, and equitable allocation of service areas, ultimately aiming to enhance the overall quality of life for children and families [6]. Further longitudinal research is recommended to assess the sustained clinical impacts and scalability of this service model. Future studies should investigate the long-term effects of the application on child growth, developmental milestones, and broader health indicators beyond caregiver satisfaction. Concurrently, periodic evaluations of the application’s workflows and user-facing forms are essential to ensure they remain accessible, contextually appropriate, and user-friendly for diverse caregiver populations.

## 5. Conclusions

The Smart Well Child Center application demonstrated significant improvements in healthcare service efficiency and caregiver satisfaction within the Pediatric Outpatient Department of Suranaree University of Technology Hospital. The integration of digital innovations effectively addressed critical service delivery challenges. High functionality and usability scores, coupled with sustained increases in caregiver satisfaction, underscore the application’s potential to enhance the accessibility and efficiency of pediatric healthcare services. Future initiatives should prioritize the expansion of the application to broader healthcare settings, the ongoing evaluation of long-term clinical outcomes, and the mitigation of digital literacy barriers among users. Additionally, further research is warranted to assess the scalability and cost-effectiveness of telehealth integration within routine pediatric care models.

## Figures and Tables

**Figure 1 healthcare-13-01676-f001:**
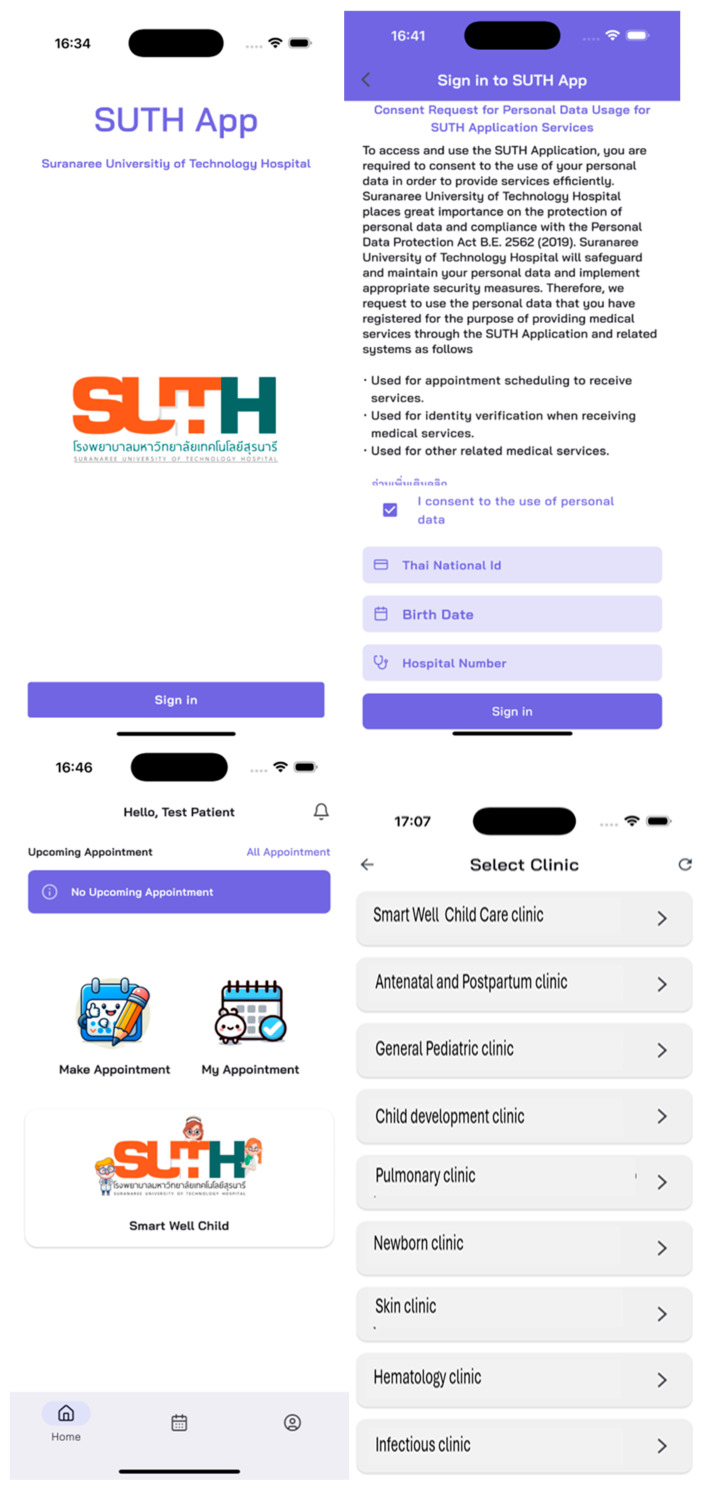
The Smart Well Child Center application features: (**a**) appointment scheduling; download application from app. Store and log in with personal ID, date of birth and hospital number. (**b1**) Smart Well Child information form for appointed patents; pre-doctor; self-assessments for caregivers, (**b2**) Smart Well Child information form for appointed patents; Smart Well Child information form for appointed patents; doctor note and post-doctor (Doctor’s advice), information for care givers; (**c1**) Line official for chatbot-based health education and telemedicine services; admins create the content for broadcasts and individual telemedicine (**c2**) Accessibility; QR code or line add friend link or click at SUTH smart well child care picture in the application. (**c3**) the main of line official; Education and chat bot for pregnancy and lactation, child development, newborn, growth and nutrition, Q&A, and return to main menu of Smart well child center application.

**Figure 2 healthcare-13-01676-f002:**
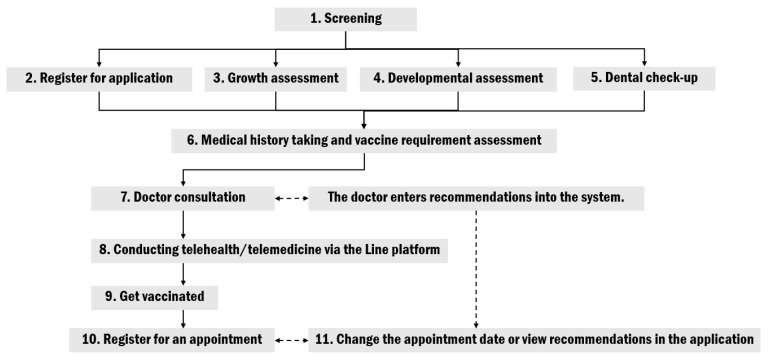
Flowchart of the service procedures.

**Table 1 healthcare-13-01676-t001:** Personal information of child caregivers and child.

Personal Information		Number	%
**Child Caregivers**			
Gender	Male	15	17.6
	Female	70	82.4
Age	Under 20 years	1	1.2
	20–30 years	32	37.6
	31–40 years	40	47.1
	41–50 years	10	11.8
	51–60 years	2	2.4
	Over 60 years	-	-
Occupation	Government official	17	20.0
	State enterprise employee	3	3.5
	Employee/staff of a company/store	24	28.2
	Business owner/trader	15	17.6
	Freelancer/daily wage worker	8	9.4
	Others	18	21.2
Relationship	Father	14	16.5
	Mother	66	77.6
	Uncle or Aunt	1	1.2
	Grandfather or Grandmother	4	4.7
Hometown	Nakhon Ratchasima	78	91.8
	Others	7	8.2
Number of Visits	First time	4	4.7
	2 times	21	24.7
	3 times or over	60	70.6
**Child**			
Gender	Male	54	63.5
	Female	31	36.5
Percentage of weight	Obese	20	23.5
for age (%: W/A)	Overweight	3	3.5
	Normal weight	53	62.4
	Slightly underweight	3	3.5
	Underweight	6	7.1
	Mean ± SD: 125.65 ± 64.27		
Percent by height	Tall	18	21.2
for age (%: H/A)	Moderately tall	2	2.4
	Normal	54	63.5
	Slightly short	3	3.5
	Short	8	9.4
	Mean ± SD: 106.8 ± 21.16		
Percent weight	Severely obese	5	6.0
for height (%: W/H)	Obese	6	7.1
	Overweight	2	2.4
	Well-proportioned	63	75.0
	Thinness	4	4.8
	Severe thinness	4	4.8
	Mean ± SD: 101.97 ± 20.45		
Bottle feeding	Not used	25	29.4
	Used	60	70.6
Gross motor skills	Age-appropriate	59	98.3
	Suspected delay	1	1.7
	Delayed	-	-
Fine motor skills	Age-appropriate	59	98.3
	Suspected delay	-	-
	Delayed	1	1.7
Emotional and social	Age-appropriate	57	95.5
	Suspected delay	3	5.0
	Delayed	-	-
Language	Age-appropriate	57	95.5
	Suspected delay	-	-
	Delayed	3	5.0
Dental check-up	Normal	60	100.0
	Tooth decay	-	-
Oral care	Fluoride coating	57	95.0
	Referred to dentistry	3	5.0

**Table 2 healthcare-13-01676-t002:** Performance evaluation of the Smart Well Child Center application.

Performance Domain	Mean ± SD	Levels
Function requirement test	4.27 ± 0.24	Highest
Function test	4.11 ± 0.31	High
Usability test	4.00 ± 0.42	High
Security test	3.92 ± 0.27	High
Overall performance	4.06 ± 0.77	High

**Table 3 healthcare-13-01676-t003:** Satisfaction with healthcare services of child caregivers.

Satisfaction with Healthcare Services	Times	Mean ± SD	Levels	Repeated Measures ANOVA or Paired *t*-Test	Times			Cohen’s d
					Before	After Use	After Use 1 Month	
Ease of access to services	Before	3.29 ± 0.56	Moderate	F = 313.274 **				0.502
	After use	4.58 ± 0.48	Highest		−1.297 **			
	After use 1 mo.	4.60 ± 0.40	Highest		−1.315 **	0.018		
Quality of medical care	Before	3.27 ± 0.42	Moderate	F = 326.162 **				0.717
	After use	4.57 ± 0.51	Highest		−1.300 **			
	After use 1 mo.	4.66 ± 0.52	Highest		−1.388 **	0.088		
Service timeliness	Before	3.26 ± 0.69	Moderate	F = 128.708 **				0.555
	After use	4.55 ± 0.58	Highest		−1.294 **			
	After use 1 mo.	4.54 ± 0.71	Highest		−1.282 **	−0.012		
Medical equipment and general environment	Before	3.58 ± 0.50	High	F = 123.906 **				0.505
	After use	4.52 ± 0.56	Highest		−0.953 **			
	After use 1 mo.	4.58 ± 0.58	Highest		−1.012 **	0.059		
Smart Well Child Care service and application	Before	-	-	t = 0.206				0.023
	After use	4.54 ± 0.55	Highest					
	After use 1 mo.	4.56 ± 0.59	Highest			−0.017		
Overall satisfaction	Before	3.35 ± 0.41	Moderate	F = 439.797 **				0.606
	After use	4.58 ± 0.57	Highest		−1.235 **			
	After use 1 mo.	4.59 ± 0.33	Highest		−1.249 **	0.014		

** Statistical significance at the level 0.01.

## Data Availability

The data presented in this study are available on request from the corresponding author due to privacy.

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
