# Peer review of "Enhancing Pediatric Outpatient Medical Services Through the Implementation of the Smart Well Child Center Application"

_healthcare, 2025, doi:10.3390/healthcare13141676_

Round 1
Reviewer 1 Report
Comments and Suggestions for Authors
The authors provide insight into experiences, developing and implementing the Smart Well Child Center App, thus aiming to enhance medical outpatient services as stated in the title.
It seems to be a large-scale project, requiring much work designing, implementing and evaluating the application designed. As stated in the manuscript, the application performed very well, when evaluated by users, thus emphasizing the important role of such an app, in outreach healthcare.
However, there are several issues to address, to prepare the manuscript for publication:
- In the abstract and introduction general, it would ease the reading and understanding of the manuscript as a research article, if an actual hypothesis or aim was supplied. If there is no research question or hypothesis, it seems the findings could just be reported as a descriptive report, rather than an academic article. Adding to this, describing the aim for monitoring or measuring the impact of the application on a user/care giver AND healthcare professional basis, moving forward would interest readers and enhance the significance of the study.
- In the introduction, the Well-child care (WCC) services are presented, it would enhance the understanding if comprehensive information on WCC in the region where the study and evaluation is set, furthermore taking into notion if the references from earlier studies are applicable in regional settings as described.
- Some references provided on eHealth is more the 10 years old (ref 7), which seems to be very “old” when evaluation a digital framework. There may be newer studies available to provide more recent findings, carrying this study into the 2020’ies, this is especially of interest when examining health- and e-literacy among users?
- Reference 10 is a bachelor thesis, if used as a reference the main findings and data should be provided, and if no novel findings are presented in the thesis, please refer to the original references?
- The section provided in line 101-110 is very well written and very important, you may consider shortening parts off the introduction to emphasize this very important section?
- In materials and methods, it is stated that the study concludes in July 2025, which is in the future? Stating the obvious, this should be changed, as you cannot evaluate future happening. If it is a matter of ongoing data collection, you must state this.
- Introducing the elements in the study design, the description of the planning phase is very swift, considering the massive work needed to develop and implement a health application, you might want to consider adding more information on the process? Maybe even a visual guide or timeline, to introduce the phases?
- In the results section, dividing the text into smaller sections would improve readability. Furthermore, I would recommend moving the section in lines 257-273 from the discussion into the results section, as it describes the features of the application, thus comprising the results from the design phase. It is very difficult to report such results; however, it seems that is what you set out to do?
- Table 1: To enhance clarity, please include a table legend defining all abbreviations.
- The quality of figures is very poor, in all elements provided in figure 1. Enlarging or cropping the snapshots may suffice, if full-sized figures are added in the supplemental material? Otherwise, you may consider adding a QR-code enabling readers to access the actual tool?
- The photos in figure 2 serves no purpose in the current quality, as it does not underscore the changes in flow within the clinic. It may be beneficial to provide a visual flowchart instead.
- In the discussion, as stated earlier in point 8, moving the first section to results would be advised. Furthermore, it is critical within the discussion, to comment on possible regional differences regarding wealth, health, literacy and similar socioeconomic differences, when comparing to results from other settings. Ref 14 is from the US in 2014, Ref 23 is from 2023, more recent studies and literature must be available.
- On a general note, the manuscript would benefit from minimizing repeated statements that do not add new information, for instance, in the conclusion line 358-360, where is reiterated once more “including appointment scheduling, pre-consultation…”, as this is presented several times earlier in the introduction, methods and discussion.
With modifications accordingly, the manuscript may be suited for publication.
Comments on the Quality of English Language
Author Response
Comments 1: In the abstract and introduction general, it would ease the reading and understanding of the manuscript as a research article, if an actual hypothesis or aim was supplied. If there is no research question or hypothesis, it seems the findings could just be reported as a descriptive report, rather than an academic article. Adding to this, describing the aim for monitoring or measuring the impact of the application on a user/care giver AND healthcare professional basis, moving forward would interest readers and enhance the significance of the study. |
Response 1: We have revised both the abstract and the introduction to explicitly state the study's primary aim: to evaluate the development, pilot implementation, and preliminary outcomes of the Smart Well Child Center application in improving caregiver engagement and streamlining pediatric service delivery. |
Comments 2: In the introduction, the Well-child care (WCC) services are presented, it would enhance the understanding if comprehensive information on WCC in the region where the study and evaluation is set, furthermore taking into notion if the references from earlier studies are applicable in regional settings as described. |
Response 2: Thank you for this insightful comment. In response, we have revised the introduction to include comprehensive information about the structure and delivery of Well-Child Care (WCC) services in Thailand, where the study is situated. We also address common challenges encountered in rural and underserved areas, such as limited access to developmental screening and follow-up services. Furthermore, we have included a critical discussion regarding the applicability of prior studies primarily from high-income settings to the Thai context. This addition emphasizes the need to interpret external findings with consideration of local healthcare infrastructure, workforce capacity, and caregiver demographics. |
Comments 3: Some references provided on eHealth is more the 10 years old (ref 7), which seems to be very “old” when evaluation a digital framework. There may be newer studies available to provide more recent findings, carrying this study into the 2020’ies, this is especially of interest when examining health- and e-literacy among users? |
Response 3: We have updated the references to include more recent and relevant studies published in 2021 and 2024, which are now cited as References 8 to 10. |
Comments 4: Reference 10 is a bachelor thesis, if used as a reference the main findings and data should be provided, and if no novel findings are presented in the thesis, please refer to the original references? |
Response 4: Upon review, we found that the bachelor thesis cited as Reference 10 did not present novel findings but instead summarized information from existing literature. As such, we have replaced this reference with the original sources cited in the thesis |
Comments 5: The section provided in line 101-110 is very well written and very important, you may consider shortening parts off the introduction to emphasize this very important section? |
Response 5: We have revised and streamlined the earlier part of the introduction to improve clarity and reduce redundancy. This adjustment allows the key section in lines 101–110—highlighting the research gap and rationale for the study—to stand out more prominently. |
Comments 6: In materials and methods, it is stated that the study concludes in July 2025, which is in the future? Stating the obvious, this should be changed, as you cannot evaluate future happening. If it is a matter of ongoing data collection, you must state this. |
Response 6: We acknowledge the confusion caused by the phrasing of the study timeline. To clarify, the study period spans from July 2023 to January 2025. We have revised the text in the Materials and Methods section. |
Comments 7: Introducing the elements in the study design, the description of the planning phase is very swift, considering the massive work needed to develop and implement a health application, you might want to consider adding more information on the process? Maybe even a visual guide or timeline, to introduce the phases? |
Response 7: We agree that the development and implementation of a health application involve a substantial amount of planning and coordination, which warrants more detailed explanation. In response, we have expanded the description of the planning phase to provide a clearer account of stakeholder involvement, system requirement gathering, and initial technical assessments. |
Comments 8: In the results section, dividing the text into smaller sections would improve readability. Furthermore, I would recommend moving the section in lines 257-273 from the discussion into the results section, as it describes the features of the application, thus comprising the results from the design phase. It is very difficult to report such results; however, it seems that is what you set out to do? |
Response 8: We have revised the Results section to include distinct subheadings that correspond to key components of the study Additionally, we have moved the content previously found in lines 257–273 from the Discussion to the Results section. |
Comments 9: Table 1: To enhance clarity, please include a table legend defining all abbreviations. |
Response 9: We have revised Table 1 to include a clear and comprehensive legend that defines all abbreviations used. This addition aims to enhance the clarity and readability of the table for all readers. |
Comments 10: The quality of figures is very poor, in all elements provided in figure 1. Enlarging or cropping the snapshots may suffice, if full-sized figures are added in the supplemental material? Otherwise, you may consider adding a QR-code enabling readers to access the actual tool? |
Response 10: We have improved the figure quality by enlarging and cropping the key snapshots to focus on essential interface elements. To further enhance accessibility, we have also added a QR code linking to a secure demonstration version of the application. |
Comments 11: The photos in figure 2 serves no purpose in the current quality, as it does not underscore the changes in flow within the clinic. It may be beneficial to provide a visual flowchart instead. |
Response 11: We have replaced the original images with a visual flowchart that illustrates the restructured service layout and patient journey across the functional zones. |
Comments 12: In the discussion, as stated earlier in point 8, moving the first section to results would be advised. Furthermore, it is critical within the discussion, to comment on possible regional differences regarding wealth, health, literacy and similar socioeconomic differences, when comparing to results from other settings. Ref 14 is from the US in 2014, Ref 23 is from 2023, more recent studies and literature must be available. |
Response 12: We have moved the first section of the discussion to the results section to maintain a clearer distinction between data presentation and interpretation. Additionally, we have expanded the discussion to address regional socioeconomic differences—including disparities in income, health system infrastructure, and caregiver health and digital literacy—which are important when comparing our findings with those from other countries. We also conducted a thorough literature update and replaced Ref 14 and Ref 23 with a more recent and contextually relevant study. |
Comments 13: On a general note, the manuscript would benefit from minimizing repeated statements that do not add new information, for instance, in the conclusion line 358-360, where is reiterated once more “including appointment scheduling, pre-consultation…”, as this is presented several times earlier in the introduction, methods and discussion. |
Response 13: We acknowledge the redundancy noted in the conclusion and agree that minimizing repetition enhances the manuscript's overall clarity and conciseness. In response, we have revised the conclusion to remove repeated details and instead focused on summarizing the key findings and implications more succinctly. |
4. Response to Comments on the Quality of English Language |
Point 1: The English could be improved to more clearly express the research. |
Response 1: We have carefully reviewed the manuscript for language clarity, grammar, and formatting. Minor revisions were made throughout the text to enhance readability and ensure consistency with academic writing standards. We appreciate your suggestion, which helped improve the overall presentation of the manuscript. |
5. Additional clarifications |
Point 1: Ensure all references are relevant to the content of the manuscript. |
Response 1: We have thoroughly reviewed and updated all references to ensure they are directly relevant to the manuscript's content. In addition, we used EndNote to manage the references and formatted them according to the MDPI citation style. |

Reviewer 2 Report
Comments and Suggestions for Authors
Dear Authors
Thank you for the opportunity to review your paper, please, see below few comments you might consider or clarify:
The study utilizes a one-group pre-post design without a control or comparison group. Despite this, the manuscript frequently uses language that suggests causal inference (e.g., “improved caregiver satisfaction,” “enhanced service efficiency”). The authors are encouraged to revise the language throughout the manuscript to reflect the observational and descriptive nature of the study design, avoiding terminology that implies causality.
Several statistical values presented in Table 3 appear implausible. For example, F-values such as “F = 23,352.01” and “F = 18,577.07” are unlikely given the reported sample size. Additionally, standard deviations such as ±1.10 for mean scores on a 1–5 scale appear inconsistent with the reported averages. The authors should carefully review the statistical outputs and consult with a qualified statistician. All p-values, F-values, and standard deviations should be verified, recalculated if needed, and corrected to ensure accurate interpretation.
The caregiver satisfaction assessment (Table 3) and performance evaluation by experts (Table 2) are presented without evidence of prior validation or reliability testing. It is recommended that the authors report internal consistency metrics (e.g., Cronbach’s alpha) or reference the validation history of the instruments used. If the tools were newly developed or adapted, this should be explicitly stated along with an explanation of their development and testing process.
In Table 1, some key findings—such as the combined proportion of children categorized as “slightly short” or “short” (~13%) and a notable obesity prevalence (23.5%)—are underexplored in the results and discussion sections. The authors should consider integrating these descriptive data more meaningfully into the interpretation of findings. Discussing their clinical relevance would strengthen the manuscript and provide a more comprehensive understanding of the population served.
The abstract and results sections report an overall application performance score of 4.06 ± 0.77. However, Table 2 shows that individual domain scores (all >3.92) would likely result in a higher mean if averaged. The authors should clarify whether the 4.06 value represents a separate data source (e.g., caregiver satisfaction rather than expert evaluation), or correct the reported value if necessary.
Extremely large F-values (e.g., F = 23,352.01) are reported in Table 3, which appears to be a miscalculation or reporting error. The authors are advised to review the analytical procedures, confirm that assumptions for repeated measures ANOVA were met, and re-report the results using correct statistical parameters. Including effect sizes such as Cohen’s d or η² would enhance interpretability.
The manuscript does not specify how the 85 child caregivers were selected. There is no mention of the sampling technique (e.g., random, convenience, consecutive) or inclusion/exclusion criteria.
The authors should clearly describe the sampling strategy, including recruitment methods, eligibility criteria, and any efforts taken to minimize selection bias.
The stated project timeline is July 2023 to July 2025, but the results and post-intervention evaluations appear to be already completed.It is suggested that the authors clarify whether the manuscript reflects an interim analysis or a pilot phase. Updating the timeline or specifying the actual period of data collection would improve clarity.
Best wishes
Author Response
Comments 1: The study utilizes a one-group pre-post design without a control or comparison group. Despite this, the manuscript frequently uses language that suggests causal inference (e.g., “improved caregiver satisfaction,” “enhanced service efficiency”). The authors are encouraged to revise the language throughout the manuscript to reflect the observational and descriptive nature of the study design, avoiding terminology that implies causality. |
Response 1: We carefully reviewed and revised the manuscript to replace terminology suggesting causality with more appropriate, observational language. Terms such as “improved,” “enhanced,” and “demonstrated” have been modified to phrases like “was associated with,” “suggested,” “perceived,” or “may support,” to better reflect the descriptive nature of the findings. |
Comments 2: Several statistical values presented in Table 3 appear implausible. For example, F-values such as “F = 23,352.01” and “F = 18,577.07” are unlikely given the reported sample size. Additionally, standard deviations such as ±1.10 for mean scores on a 1–5 scale appear inconsistent with the reported averages. The authors should carefully review the statistical outputs and consult with a qualified statistician. All p-values, F-values, and standard deviations should be verified, recalculated if needed, and corrected to ensure accurate interpretation. |
Response 2: We reviewed the statistical outputs in Table 3. The previously reported F-values and standard deviations were found to be incorrectly recorded. We have now corrected these values based on accurate output from repeated measures ANOVA, consistent with the sample size and the scale used (1–5 Likert scale). |
Comments 3: The caregiver satisfaction assessment (Table 3) and performance evaluation by experts (Table 2) are presented without evidence of prior validation or reliability testing. It is recommended that the authors report internal consistency metrics (e.g., Cronbach’s alpha) or reference the validation history of the instruments used. If the tools were newly developed or adapted, this should be explicitly stated along with an explanation of their development and testing process. |
Response 3: We have clarified that both the caregiver satisfaction assessment and the expert performance evaluation forms underwent content validity assessment through expert review, and the Index of Consistency (IOC) was calculated. The instruments were then piloted, and internal consistency was evaluated using Cronbach’s alpha. The caregiver satisfaction assessment yielded a reliability score of 0.95, and the expert evaluation form yielded 0.92. |
Comments 4: In Table 1, some key findings—such as the combined proportion of children categorized as “slightly short” or “short” (~13%) and a notable obesity prevalence (23.5%)—are underexplored in the results and discussion sections. The authors should consider integrating these descriptive data more meaningfully into the interpretation of findings. Discussing their clinical relevance would strengthen the manuscript and provide a more comprehensive understanding of the population served. |
Response 4: We have revised the Discussion section to incorporate and elaborate on the descriptive findings reported in Table 1. These data were contextualized with national statistics from the 4th National Health Examination Survey (NHES IV) and linked to the need for early and targeted nutritional and developmental interventions. We also expanded the discussion on the implications of developmental delays identified through DSPM screening. This integration strengthens the interpretation of our findings and provides a clearer understanding of the healthcare needs within the population served by the Smart Well Child Center. |
Comments 5: The abstract and results sections report an overall application performance score of 4.06 ± 0.77. However, Table 2 shows that individual domain scores (all >3.92) would likely result in a higher mean if averaged. The authors should clarify whether the 4.06 value represents a separate data source (e.g., caregiver satisfaction rather than expert evaluation), or correct the reported value if necessary. |
Response 5: We acknowledge the confusion caused by the reporting of the overall performance score. To clarify, the score of 4.06 ± 0.77 presented in the abstract and results refers to the expert performance evaluation, not caregiver satisfaction. We have corrected this value in both the abstract and results sections, and the revised data are now accurately reflected in Table 2. |
Comments 6: Extremely large F-values (e.g., F = 23,352.01) are reported in Table 3, which appears to be a miscalculation or reporting error. The authors are advised to review the analytical procedures, confirm that assumptions for repeated measures ANOVA were met, and re-report the results using correct statistical parameters. Including effect sizes such as Cohen’s d or η² would enhance interpretability. |
Response 6: We have reviewed the statistical outputs and confirm that the previously reported F-values in Table 3. These values have now been corrected to reflect the accurate results of the repeated measures ANOVA tests. Furthermore, we have included effect size measures (η²) to improve the interpretability of the findings. The corrected F-values and corresponding effect sizes have been updated in Table 3 and referenced in the revised Results section. |
Comments 7: The manuscript does not specify how the 85 child caregivers were selected. There is no mention of the sampling technique (e.g., random, convenience, consecutive) or inclusion/exclusion criteria. |
Response 7: We have revised the manuscript to include details regarding the participant selection process. Specifically, we clarify that a simple random sampling technique was used. Participants included primary caregivers of children under six years of age who had accessed the Smart Well Child Center services at least once during the study period. Caregivers were eligible if they were aged 18 or older and able to provide informed consent. Those with communication barriers or who declined to participate were excluded. |
Comments 8: The authors should clearly describe the sampling strategy, including recruitment methods, eligibility criteria, and any efforts taken to minimize selection bias. |
Response 8: We have revised the manuscript to include a detailed description of the sampling strategy. Specifically, we used a simple random sampling method to recruit 85 eligible caregivers from the pediatric outpatient department. Inclusion criteria were caregivers aged 18 years or older who were responsible for children under six years of age and had used the Smart Well Child Center services at least once during the study period. Caregivers with communication barriers or who declined to provide informed consent were excluded. To minimize selection bias, recruitment was conducted across different clinic days and time slots to capture a representative sample of users. |
Comments 9: The stated project timeline is July 2023 to July 2025, but the results and post-intervention evaluations appear to be already completed. It is suggested that the authors clarify whether the manuscript reflects an interim analysis or a pilot phase. Updating the timeline or specifying the actual period of data collection would improve clarity. |
Response 9: We have revised the Study Design section to indicate that the manuscript presents findings from a pilot implementation phase, conducted between September to November 2024, as part of the broader project scheduled from July 2023 to January 2025. |
4. Response to Comments on the Quality of English Language |
Point 1: The English is fine and does not require any improvement. |
Response 1: We have carefully reviewed the manuscript for language clarity, grammar, and formatting. Minor revisions were made throughout the text to enhance readability and ensure consistency with academic writing standards. We appreciate your suggestion, which helped improve the overall presentation of the manuscript. |
5. Additional clarifications |
Point 1: Ensure all references are relevant to the content of the manuscript. |
Response 1: We have thoroughly reviewed and updated all references to ensure they are directly relevant to the manuscript's content. |

Reviewer 3 Report
Comments and Suggestions for Authors
It was a real pleasure to read your manuscript. Your work addresses a very relevant and practical issue, and I was particularly impressed by the thoughtful integration of digital tools into the pediatric outpatient setting. The Smart Well Child Center application seems like a promising step forward in making healthcare more accessible and parent-friendly. I have only a few minor suggestions that might help further improve the clarity and impact of your paper: 1) In the Discussion, it could be useful to more clearly acknowledge certain methodological limitations -such as the lack of a control group and the short follow-up period. 2)If possible, including even brief insights into caregivers' experiences or feedback (beyond quantitative satisfaction scores) would enrich the reader’s understanding of the app’s real-world impact. 3)Finally, a light language and formatting check would help polish the already clear writing. These are small points in an otherwise strong and well-structured manuscript. I truly hope this valuable work receives the attention it deserves. Warm wishes and best of luck with your revisions.
Author Response
Comments 1: In the Discussion, it could be useful to more clearly acknowledge certain methodological limitations -such as the lack of a control group and the short follow-up period. |
Response 1: We have revised the Discussion section to explicitly acknowledge key methodological limitations, including the absence of a control group and the short follow-up period. |
Comments 2: If possible, including even brief insights into caregivers' experiences or feedback (beyond quantitative satisfaction scores) would enrich the reader’s understanding of the app’s real-world impact. |
Response 2: We have incorporated a summary of informal caregiver feedback collected during the follow-up period. These insights highlight the perceived benefits of the application, including increased convenience in appointment management, reduced anxiety from pre-consultation self-assessments, and improved access through teleconsultation features. |
Comments 3: Finally, a light language and formatting check would help polish the already clear writing. |
Response 3: We have carefully reviewed the manuscript for language clarity, grammar, and formatting. Minor revisions were made throughout the text to enhance readability and ensure consistency with academic writing standards. We appreciate your suggestion, which helped improve the overall presentation of the manuscript. |
4. Response to Comments on the Quality of English Language |
Point 1: The English could be improved to more clearly express the research. |
Response 1: We have carefully reviewed the manuscript for language clarity, grammar, and formatting. Minor revisions were made throughout the text to enhance readability and ensure consistency with academic writing standards. We appreciate your suggestion, which helped improve the overall presentation of the manuscript. |
5. Additional clarifications |
Point 1: Ensure all references are relevant to the content of the manuscript. |
Response 1: We have thoroughly reviewed and updated all references to ensure they are directly relevant to the manuscript's content. |

Reviewer 4 Report
Comments and Suggestions for Authors
The manuscript titled “Enhancing Pediatric Outpatient Medical Services Through the Implementation of the Smart Well Child Center Application” presents a timely and relevant digital health intervention tailored for pediatric outpatient services in Thailand. The study is well-structured, employing the System Development Life Cycle (SDLC) framework and combining technical development with real-world evaluation. It addresses common barriers such as long wait times, limited accessibility, and caregiver dissatisfaction by integrating appointment scheduling, teleconsultation, and educational tools into a single mobile platform. However, several issues should be addressed before publication.
First, while the authors note a lack of integrated systems in Thailand, the novelty of the application is not convincingly distinguished from existing global eHealth solutions. The manuscript would benefit from a stronger emphasis on local challenges—such as digital literacy and infrastructure—and how this intervention uniquely addresses those gaps. Second, the limited sample size (85 caregivers) drawn from a single site restricts the generalizability of the findings. This limitation is not sufficiently discussed, nor is there a plan for future scaling. Third, the absence of a control group raises concerns about possible bias in caregiver satisfaction reporting, which may be inflated due to the novelty effect. The evaluation also lacks detail on data privacy protections, an important consideration given the sensitivity of pediatric data.
Furthermore, while performance evaluation scores are promising, the statistical analysis would be more robust with the inclusion of effect sizes and confidence intervals. Descriptions of certain app features, such as the SUTH form and chatbot logic, remain too general and would benefit from additional technical detail or illustrative figures. Lastly, the discussion occasionally reiterates results rather than critically engaging with limitations, implementation risks, or the broader context of pediatric digital health equity.
In summary, this is a promising and relevant contribution to healthcare service innovation. With revisions that improve the articulation of its originality, methodological transparency, and limitations, the article could serve as a strong model for pediatric eHealth development in middle-income countries. A recommendation for minor revision is warranted.
Author Response
Comments 1: While the authors note a lack of integrated systems in Thailand, the novelty of the application is not convincingly distinguished from existing global eHealth solutions. The manuscript would benefit from a stronger emphasis on local challenges—such as digital literacy and infrastructure—and how this intervention uniquely addresses those gaps. |
Response 1: We have revised the manuscript to emphasize how the Smart Well Child Center application was designed to address local challenges, including limitations in digital literacy, language, and infrastructure. Specifically, we expanded the section describing the Smart Well Child novel SUTH form to highlight its structured, three-step format which improve health literacy (Pre-doctor, Doctor note, Post-doctor directed to the SUTH application), which was developed based on AAP guidelines and refined through consensus among local pediatricians. This form was adapted for ease of use by Thai caregivers, many of whom face barriers related to health literacy and limited digital experience. |
Comments 2: The limited sample size (85 caregivers) drawn from a single site restricts the generalizability of the findings. This limitation is not sufficiently discussed, nor is there a plan for future scaling. |
Response 2: We have expanded the limitations section of the Discussion to explicitly acknowledge this issue and its implications. |
Comments 3: The absence of a control group raises concerns about possible bias in caregiver satisfaction reporting, which may be inflated due to the novelty effect. The evaluation also lacks detail on data privacy protections, an important consideration given the sensitivity of pediatric data. |
Response 3: We have revised the Discussion section to explicitly acknowledge key methodological limitations, including the absence of a control group and the short follow-up period. Additionally, we have added a description of data privacy and security measures implemented during the study, including encryption protocols, user authentication processes, and compliance with institutional data protection guidelines. These details have been included in the Methods section |
Comments 4: While performance evaluation scores are promising, the statistical analysis would be more robust with the inclusion of effect sizes and confidence intervals. Descriptions of certain app features, such as the SUTH form and chatbot logic, remain too general and would benefit from additional technical detail or illustrative figures. Lastly, the discussion occasionally reiterates results rather than critically engaging with limitations, implementation risks, or the broader context of pediatric digital health equity. |
Response 4: We have now included effect size metrics (e.g., Cohen’s d). Additionally, we have expanded the description of the SUTH form and chatbot logic in the Methods section, providing a more detailed explanation of the form’s structure (pre-doctor, doctor, and post-doctor components). Furthermore, we have revised the Discussion section to reduce repetition and have added a more critical analysis of implementation challenges, potential risks (e.g., over-reliance on digital systems, disparities in digital access), and the relevance of digital health equity in pediatric care. |
4. Response to Comments on the Quality of English Language |
Response 1: We have carefully reviewed the manuscript for language clarity, grammar, and formatting. Minor revisions were made throughout the text to enhance readability and ensure consistency with academic writing standards. We appreciate your suggestion, which helped improve the overall presentation of the manuscript. |
5. Additional clarifications |
Point 1: Ensure all references are relevant to the content of the manuscript. |
Response 1: We have thoroughly reviewed and updated all references to ensure they are directly relevant to the manuscript's content. |

Round 2
Reviewer 1 Report
Comments and Suggestions for Authors
Thank you for the revised manuscript. My concerns have been met.
I have no further comments.
Reviewer 2 Report
Comments and Suggestions for Authors
Thank you for addressing the comments